# How heat propagates in liquid ³He

**Kamran Behnia** [1] ✉ **& Kostya Trachenko** [2]

In Landau's Fermi liquid picture, transport is governed by scattering between quasi-particles. The normal liquid ³He conforms to this picture but only at very low temperature. Here, we show that the deviation from the standard behavior is concomitant with the fermion-fermion scattering time falling below the Planckian time, $\frac{\hbar}{k_B T}$ and the thermal diffusivity of this quantum liquid is bounded by a minimum set by fundamental physical constants and observed in classical liquids. This points to collective excitations (a sound mode) as carriers of heat. We propose that this mode has a wave-vector of $2k_F$ and a mean free path equal to the de Broglie thermal length. This would provide an additional conducting channel with a $T^{1/2}$ temperature dependence, matching what is observed by experiments. The experimental data from 0.007 K to 3 K can be accounted for, with a margin of 10%, if thermal conductivity is the sum of two contributions: one by quasi-particles (varying as the inverse of temperature) and another by sound (following the square root of temperature).

The impact of Landau's Fermi liquid (FL) theory[1] in condensed matter physics of the twentieth century can not be exaggerated. Before its formulation, the success of Sondheimer's picture of electrons in metals as a degenerate Fermi gas was a mystery[2]. Even in a simple alkali metal such as Na, the Coulomb interaction between electrons is larger than the Fermi energy. Why wasn't this a problem for understanding the physics of metallic solids? Landau solved this mystery by proposing that a strongly interacting system of fermions can be mapped to an ideal system consisting of "quasi-particles" without interaction, or rather with an interaction weak enough to be considered as a perturbation[2,3]. His theory was inspired by the less common isotope of helium, namely ³He[4], the first experimental platform for testing the theory. Decades later, the theory was also applied to strongly correlated metals, known as heavy-fermion systems[5].

Leggett, reviewing liquid ³He[6], writes that it is "historically the first strongly interacting system of fermions of which we have been able to obtain a semi-quantitative description in the low-temperature limit." He also adds that the theory "seems to agree quantitatively with experiment only for $T \lesssim 100$ mK". This is very low compared to ~ 5 K, the degeneracy temperature of non-interacting fermions calculated with the bare mass of the ³He atoms. The properties of the ground state (and its evolution with pressure) have been the subject of numerous theoretical papers[7–14]. In contrast, the breakdown of the Fermi liquid picture at very low temperatures, earlier noted by Emery[15] and Anderson[16], ceased to be widely debated afterwards.

Here, we begin by recalling that as low as $T \approx 0.01$ K, the Fermi liquid picture does not hold. The Fermi temperature, with the mass normalization taken into account, is $T_F \sim 2$K. According to the most comprehensive set of data[17], when $\frac{T}{T_F} \approx 5 \times 10^{-3}$ thermal conductivity, $\kappa$, deviates from the expected $T^{-1}$ behavior and the extracted scattering time, $\tau_\kappa$, is no more $\propto T^{-2}$. We show that this "non-Fermi-liquid" (NFL) regime emerges when the fermion-fermion scattering time becomes comparable or shorter than the Planckian time[18,19], the time scale often invoked in the context of "strange" metallicity[20]. Remarkably, in this regime, the thermal diffusivity, $D_{th}$, of quantum liquid ³He matches the minimum empirically observed[21] and theoretically justified[22,23] in classical liquids. We conclude that collective excitations play a role in heat transport comparable to the role played by phonons in classical liquids. We find that the magnitude and the temperature dependence of the thermal conductivity can be accounted for if heat is carried by a hydrodynamic sound mode with a $2k_F$ wave-vector and a spatial evanescence set by the thermal thickness of the Fermi surface in the momentum space[24]. This phononic mechanism of heat propagation is distinguished from all those previously identified in solids and liquids, either classical or quantum. On the other hand, when the temperature becomes of the

---

¹Laboratoire de Physique et d'Étude des Matériaux, (ESPCI - CNRS - Sorbonne Université), PSL Research University, Paris, France. ²School of Physical and Chemical Sciences, Queen Mary University of London, London, UK. ✉e-mail: Kamran.Behnia@espci.fr

order of the Fermi temperature, its expression becomes another version of the Bridgman formula[25] for classical liquids.

## Results and discussion

Figure 1 reproduces figures reported by Greywall, who performed the most extensive study of thermal transport in normal liquid ³He[17]. Samples with different molar volumes correspond to different pressures in the $T = 0$ limit. As seen in Fig. 1a, thermal conductivity, $\kappa$, at low temperature is inversely proportional to temperature, as expected in the FL picture. But the temperature window for this behavior, already narrow at zero pressure, shrinks with increasing pressure. By the melting pressure, the FL regime has almost vanished. The breakdown is even more visible in Fig. 1b. It shows the temperature dependence of the inverse of of $\tau_\kappa T^2$, the scattering time extracted from thermal conductivity multiplied by the square of temperature, which should be constant in the Fermi liquid picture. A deviation is visible even at 8 mK and shoots up with increasing pressure.

The deviation from the Fermi liquid behavior was usually attributed to spin fluctuations (see for example[9]). While such a correction is expected at very low temperature, it is hard to see how they can play a role in our temperature of interest given the small amplitude of the exchange energy (See Supplementary Note 1).

In Fig. 2a, we compare the temperature dependence of $\tau_\kappa$ according to Greywall's data (Fig. 1b) with the Planckian time, $\tau_P = \frac{\hbar}{k_B T}$[18,19]. One can see that, at zero pressure, $\tau_\kappa$ becomes of the order of $\tau_P$ at $T \approx 0.1$ K. At 3 MPa, near the melting pressure, $\tau_\kappa$ falls below $\tau_P$ at $\approx 0.043$ K. As seen in Fig. 2b, which reproduces the phase diagram of ³He[4,10,14], the crossover between the FL and the NFL regions of the phase diagram is concomitant with the passage from $\tau_\kappa \gg \tau_P$ to $\tau_\kappa \lesssim \tau_P$. The possibility of a Planckian bound on dissipation is a subject hotly debated in condensed matter physics[18].

An important clue is provided by the temperature dependence of thermal diffusivity, $D_{th}$, obtained from $\kappa$[17] and specific heat[26]. As seen in Fig. 3a, it shows a minimum, both at zero pressure and at 3 MPa.

In all classical fluids, thermal diffusivity goes through a minimum at the intersection between a liquid-like regime, where it decreases with temperature, and a gas-like regime where it increases with temperature[21,27]. We illustrate this in Fig. 3b, which shows the temperature dependence of $D_{th}$ in two fluids with slightly different atomic or molecular masses, namely $H_2$ and ⁴He[27]. In all three cases, there is a minimum of thermal diffusivity, consistent with other classical liquids[21,27]. Although the minima are seen at different temperatures, the minimum $D_{th}$ has a similar amplitude: Expressed in mm² s⁻¹, $D_{min}$, is 0.063 in ³He, ~0.049 in ⁴He, and ~0.065 in $H_2$. This minimum is set by fundamental physical constants (See Supplementary Note 2).

The closeness of the minima in ³He and other classical liquids suggests an important role played by phonon-like collective excitations of ³He in heat transport as they do in classical liquids considered earlier[21]. Indeed, diffusion at high temperature is driven by the random walk of the particles. Cooling lowers the diffusion constant by decreasing the velocity and the mean free path close to interatomic separation. Below a given temperature, collective excitations (e.g., sound) begin to operate. In a quantum liquid, this process occurs along the opposite direction: the diffusion constant is dominated by quasiparticles at low temperature. With warming, the mean path decreases and approaches its minimal value as in classical liquids[21], as is seen in Fig. 3a. When thermal conductivity due to quasiparticles becomes small, the other remaining mechanism becomes important, namely conductivity due to collective excitations, sound. Using Landau's own words, "sonic excitations in the gas of quasi-particles (phonons of the "zeroth sound")"[28], are to become the main carriers of heat above this minimum.

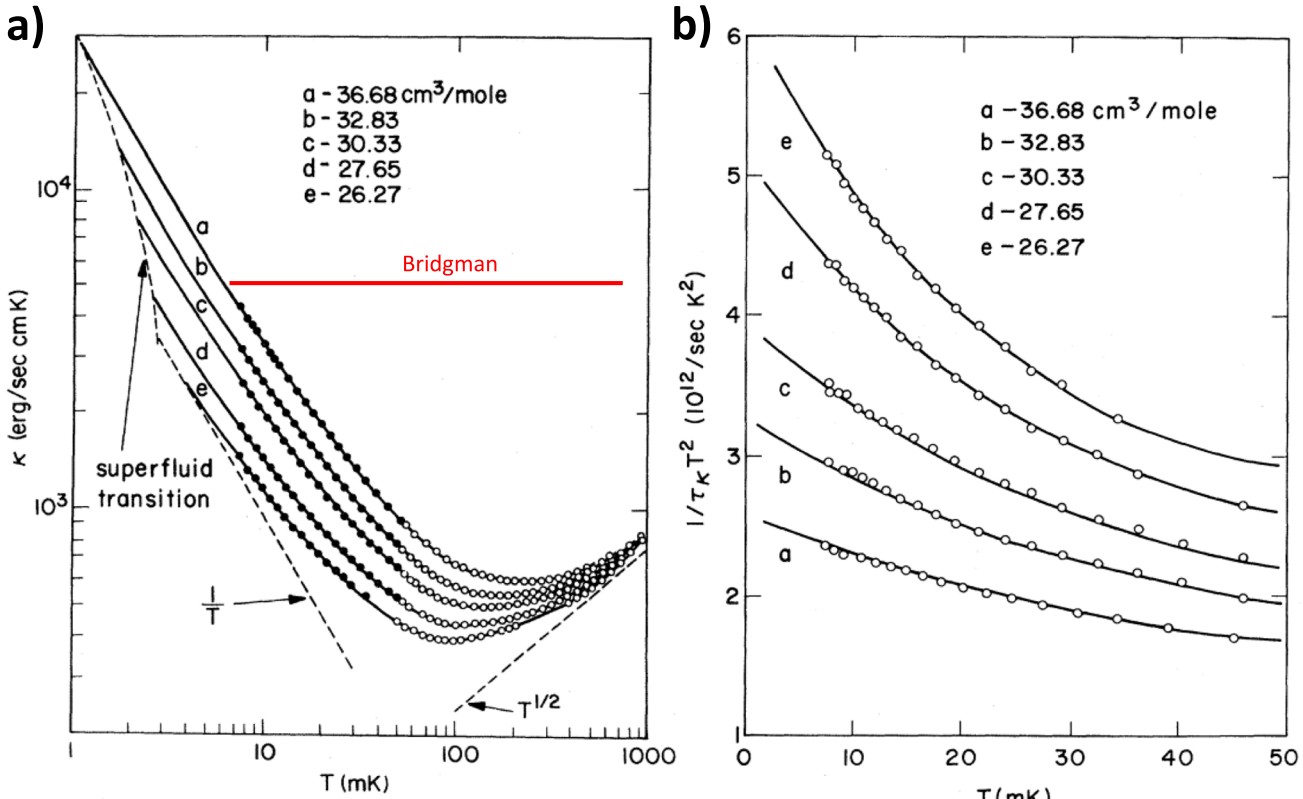

**Fig. 1 | The narrow validity of the Fermi liquid picture of the thermal conductivity in ³He. a** Thermal conductivity, $\kappa$ as a function of temperature. **b** The scattering time $\tau_\kappa$ extracted from the same data and specific heat, times the square of temperature, $T^2$. Contrary to what is expected in the standard Fermi liquid theory, $\tau_\kappa T^2$ is never constant. The figures are from Ref. 17. The horizontal red solid line in **a** represents what is expected for a classical liquid according to Bridgman's formula. Reproduced with the permission from the American Physical Society.

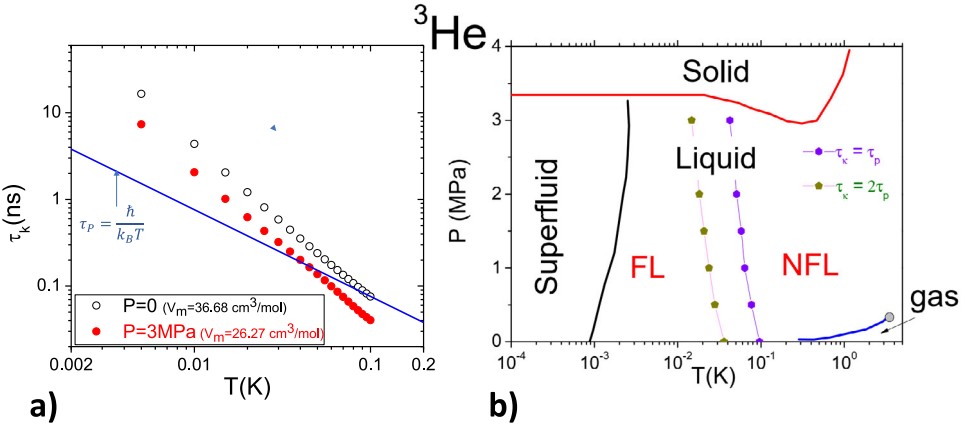

**Fig. 2 | Scattering time, Planckian bound, and the FL-NFL cross-over. a** Fermion-fermion scattering time, $\tau_\kappa$ as a function of temperature for two molar volumes[17]. The blue solid line represents the Planckian time: $\tau_P = \frac{\hbar}{k_B T}$. Note that $\tau_\kappa$ tends to fall below $\tau_P$ at sufficiently high temperature. **b** The phase diagram of $^3$He[10] and the fuzzy border between the Fermi liquid (FL) and the non-Fermi liquid (NFL) regimes. Also shown are temperatures below which $\tau_\kappa > \tau_P$ or $\tau_\kappa > 2\tau_P$.

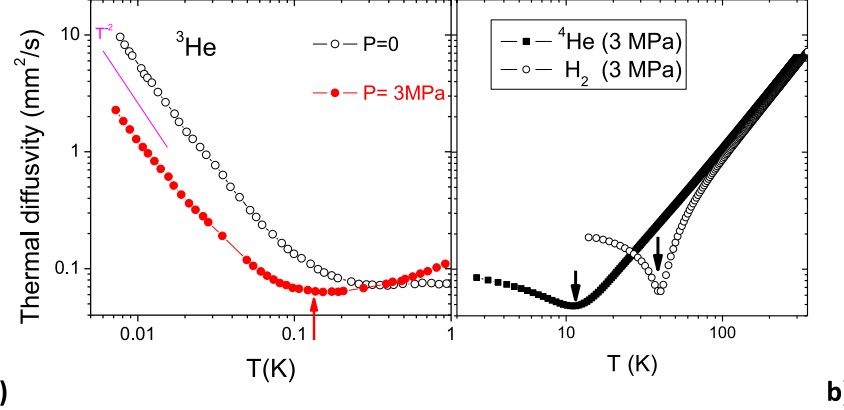

**Fig. 3 | Bounds to thermal diffusivity. a** Thermal diffusivity, $D_{th} = \kappa/C$, of $^3$He, as a function of temperature. The Fermi liquid regime ($\propto T^{-2}$) is restricted to low temperatures. It is followed by a saturation and a minimum. The curves are plotted using Greywall's thermal conductivity[17] and specific heat[26] data. **b** Thermal diffusivity as a function of temperature in two classical fluids ($H_2$ and $^4$He in the classical regime)[27]. In all three cases, $D_{th}$ has close values at the minimum, as shown by the arrows.

Collision time becomes shorter than the Planckian time when the frequency of thermally excited zero sound (which increases linearly with temperature ($\omega_{zs} = \frac{k_B T}{\hbar}$)) becomes smaller than the scattering rate (which increases quadratically with temperature). This inequality ($\omega_{zs}\tau_\kappa < 1$) means that the thermally excited zero sound is in the hydrodynamic limit, where the distinction between zero sound and first sound fades away[3]. In liquid $^3$He, this occurs at a remarkably low temperature, because $\tau_\kappa T^2 \ll \frac{\hbar E_F}{k_B^2}$ compared to electrons in metals[29,30] (see Supplementary Note 3).

As seen in Fig. 1a, Greywall observed that above 0.5 K, $\kappa \propto T^{1/2}$. Figure 4a shows the temperature dependence of $\kappa/T$ in normal liquid $^3$He. It includes Greywall's data at $T < 1$ K[17], measured at the constant molar volume of 36.68 cm$^3$/mol (corresponding to zero pressure in the low-temperature limit) and the data reported by Murphy and Meyer[31], measured at saturating vapor pressure (SVP) above 1.2 K (Fig. 4a). Despite the imperfect agreement (unsurprising given the change in the molar volume at SVP), one can see that Murphy and Meyer[31] roughly confirm Greywall's observation about the asymptotic tendency of thermal conductivity: $\kappa \propto T^{1/2}$. This temperature dependence is distinct from what is known to occur in different regimes of phonon thermal conductivity in crystals and glasses (see Table 1) and also from the Bridgman formula ($\kappa_B = r k_B n^{2/3} v_s$)[25,32–34] for classical liquids, which does not contain any temperature dependence.

To find the source of the temperature dependence of thermal conductivity, let us turn to Landauer's picture of conduction as transmission[35]. When heat is transmitted by a wave, thermal conductivity becomes[24,36,37] (See Supplementary Note 4):

$$\frac{\kappa}{T} = \frac{\pi^2}{3}\frac{k_B^2}{h}\mathcal{T} \tag{1}$$

Here, $\frac{\pi^2}{3}\frac{k_B^2}{h}$ is the quantum of thermal conductance[38]. The transmission coefficient $\mathcal{T}$ is set by the number of carrier modes and their mean free path. Its units depends on dimensions (meters in 1D, dimensionless in 2D and meters$^{-1}$ in 3D). In three dimensions, a spherical Fermi surface of radius $k_F$ contains $\frac{8\pi}{3\lambda_F^2}$ conducting modes and with a quasi-particle mean-free-path set by scattering, $\ell_{qp}$, the thermal conductivity becomes:

$$\frac{\kappa}{T}\Big|_{qp} = \frac{2\pi}{9}\frac{k_B^2}{h}k_F^2 \ell_{qp} \tag{2}$$

At very low temperature, the response to temperature gradient is dominated by quasi-particles (Fig. 4b). These are plane waves within a thermal window of the Fermi level which can carry heat. $\kappa/T$ decreases quadratically with temperature, due to the temperature dependence of $\ell_{qp}$.

To identify the collective transport mode leading to the $T^{1/2}$ temperature dependence dominant above 0.5 K, let us compare it with two other cases. In crystals, the cubic temperature dependence of phonons at low temperature reflects the temperature dependence of the volume of the Debye sphere when the mean-free-path is saturated to a constant value. In glasses, the asymptotic low-temperature dependence of thermal conductivity is close to quadratic. The slower temperature dependence $\kappa$ is due to an increase in the mean-free path with

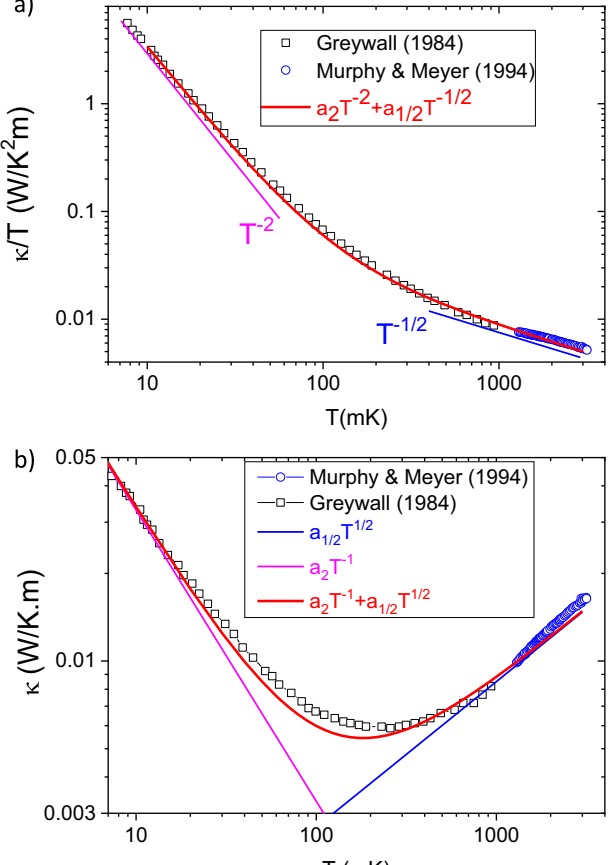

a)

b)

**Fig. 4 | Experimental data and our model. a** Thermal conductivity is divided by temperature as reported by Greywall[17] (at zero pressure) and by Murphy and Meyer[31] (at saturated vapor pressure). At very low temperatures ($T$ - 0.01 K) in the Fermi liquid regime, $\kappa/T \propto T^{-2}$ (purple line). Above 0.5 K, $\kappa/T \propto T^{-1/2}$. The blue line represents what is expected by Eq. (4), using the effective mass and the carrier density of $^3$He. The red solid line represents a fit to the experimental data in the whole temperature range assuming that $\kappa/T$ consists of the sum of $T^{-1/2}$ (sound transmission) and $T^{-2}$ term (quasi-particle transmission) terms. **b** Same data plotted for $\kappa(T)$. The maximum discrepancy between data and theory is about 10%. Also shown are the two components of the total $\kappa(T)$.

cooling. In both cases, the presence of long wavelength carriers leads to a superlinear exponent (between 2 and 3) in the temperature dependence. Our case requires a scenario circumventing the cubic temperature dependence of a Debye sphere.

A collective transmission by the whole Fermi surface will meet this requirement. This would be a sound mode with a wave-vector fixed at twice the Fermi radius (Fig. 4c). There are two reasons for distinguishing $2k_F$ as a wave vector. The first is theoretical. The Lindhard function, which quantifies the susceptibility of a fermionic gas to an external perturbation has a singularity at $q = 2k_F$[39]. The second is experimental. Inelastic X-ray scattering experiments[40,41] find that the dispersion of zero sound has a pronounced anomaly near $q = 2k_F$ (see Supplementary Note 5).

The Landauer transmission rate of such a heat-carrying mode depends on its evanescence. The wave is attenuated by the thermal fuzziness of the Fermi surface in the momentum space, which is set by the inverse of the de Broglie thermal length[24]. We can consider the latter, $\Lambda = \frac{h}{\sqrt{2\pi m^* k_B T}}$ ($m^*$ is the effective mass), as a mean free path. The number of involved states is identical to the one used to quantify the quasi-particle contribution. Replacing $\mathcal{T}$ in Eq. (1) then gives:

$$\frac{\kappa}{T}|_s = \frac{2\pi}{9}\frac{k_B^2}{h}k_F^2\Lambda \tag{3}$$

Substituting $\Lambda$ by its explicit value and the Fermi wave-vector with the particle density (with $n = \frac{k_F^3}{3\pi^2}$) leads to:

$$\kappa_s \simeq r k_B n^{2/3}\sqrt{\frac{2k_B T}{m^*}} \tag{4}$$

Here $r = \frac{\pi^{11/6}}{3^{4/3}} \approx 1.88$.

Equation (4) has two parameters, the particle density, $n$, and the effective mass, $m^*$. In Fig. 4a, it is plotted using the effective mass ($m^* = 2.7\ m_3$[26]$= 1.35 \times 10^{-26}$ kg) and the zero-pressure carrier density ($n = 1.64 \times 10^{28}$ m$^3$, corresponding to a molar volume of 36.68 cm$^3$/mol[17]) of normal liquid $^3$He. At 1 (2) K, the experimentally measured thermal conductivity is 11 (20) percent larger what is expected from equation (4).

An additional conduction channel by sound would provide an explanation for the narrow validity of the standard Fermi liquid approach. As the red line in Fig. 4a shows, at any arbitrary temperature between 0.01 K and 3 K, the experimentally measured $\kappa/T$ can be described by a sum of $T^{-2}$ and $T^{-1/2}$ terms. Fig. 4b shows the same data in a plot of $\kappa(T)$. One can see that in the whole temperature range, the total thermal conductivity can be expressed as a sum of two terms:

$$\kappa(T) = \kappa_{qp}(T) + \kappa_s(T) \tag{5}$$

where $\kappa_{qp} \propto T^{-1}$ is given by Eq. (2) and $\kappa_s \propto T^{1/2}$ is given by Eq. (3) (or equivalently by Eq. (4)).

The fit parameters used for the red curves in Fig. 4, $a_2$ and $a_{1/2}$ for $\kappa_{qp}/T = a_2 T^{-2}$ and $\kappa_s/T = a_2 T^{-1/2}$, are listed in Table 2. The table also

## Table 1 | Different cases of phonon thermal conductivity

| System | Sonic heat carriers | Temperature dependence | Mechanism | Reference |
|---|---|---|---|---|
| Crystal ($T \to 0$) | small-$q$ phonons | $\kappa \propto T^3$ | Boundary scattering | 39,47 |
| Crystal ($T \sim T_D$) | large-$q$ phonons | $\kappa \propto T^{-1}$ | Umklapp scattering | 47–49 |
| Glass ($T \to 0$) | "propagons" | $\kappa \propto T^{-2}$ | Rayleigh scattering | 50–52 |
| Glass (high T) | "diffusons" | $\kappa \propto T^0$ | Minimum mean free path | 52–54 |
| Classical liquid | large-$q$ phonons | $\kappa \propto T^0$ | Minimum mean free path | 25 |
| Quantum liquid $^3$He | $q \sim 2k_F$ phonons | $\kappa \propto T^{1/2}$ | Fermi surface thermal fuzziness | This work |

Comparison between the present case of heat propagation by collective excitations with other and better-understood regimes of phonon thermal conductivity.

compares the amplitude of $a_2$ with previous experimental[17,42] and theoretical[8,9] estimations of it.

Given the simplicity of the picture drawn above, this is a surprisingly good agreement. Let us recall that in the case of the S.V.P. data by Murphy and Meyer[31], particle density is not constant and decreases with warming. Moreover, the effective mass also changes with temperature. Finally, not only our simple model neglect any change in density and mass, but it also does not take into account a finite coupling between the two carriers of heat (particles and sound). Nevertheless, the quantitative difference between the

expected and the measured thermal conductivity does not exceed 10 percent.

Equation (4) has a striking resemblance to the Bridgman formula for classical liquids[25,32–34] (see Supplementary Note 6), with the speed of sound replaced by a group velocity of $\sqrt{\frac{2k_B T}{m}}$. This can be accounted for by noticing that the time scale for randomness in a quantum liquid is set by the ratio of the Fermi velocity to the de Broglie thermal length (and not by the ratio of the sound velocity to the inter-particle distance, as in a classical liquid or a glass). Interestingly, the two equations become similar at the classical/quantum boundary, that is when the de Broglie thermal length becomes equal to the Fermi wavelength (see Supplementary Note 6).

Figure 5 summarizes our main message. It appears that in a Fermi liquid two modes of conduction are at work. The first (Fig. 5a) has been understood for decades and is based on collisions between quasi-particles. The second, identified here, is the breathing of the whole Fermi surface (Fig. 5b). Interestingly, the two channels differ only by their respective relevant length scale, the distance between two successive collisions, and the thermal de Broglie length.

In the real space, this collective mode is presumably a visco-elastic[4,43] soft phonon with the shortest possible wavelength, which is the interatomic distance. As seen in Fig. 5c, the wavelength of the sound in question is almost identical to interatomic distance. Such a wave can be generated either by a leftward or rightward shift of all

**Table 2 | Numerical amplitudes of a₂ and a₁/₂**

| $a_2$ ($10^{-4}$ W·m$^{-1}$) | $a_{1/2}$ ($10^{-2}$ W·m$^{-1}$·K$^{-3/2}$) | Reference |
|---|---|---|
| 3.3 | 8.5 | This work (Fit in Fig. 4) |
| — | 7.6 | This work (Eq. (4): $m^* = 2.7m_3$ & $n = 1.63 \times 10^{22}$) |
| 2.9 | — | Ref. 17 (experiment) |
| 3.5 | — | Ref. 42 (experiment) |
| 3.3–5.4 | — | Ref. 9 (theory) |
| 5 | — | Ref. 8 (theory) |

The parameters of the fit to $\frac{\kappa}{T} = a_2 T^{-2} + a_{1/2} T^{-1/2}$ plotted in Fig. 4a. Also listed are $a_{1/2}$ according to Eq. (4) (with no adjustable parameter) and previous experimental and theoretical reports on the magnitude of the prefactor of the quasi-particle contribution, $a_2[\equiv \kappa T]$.

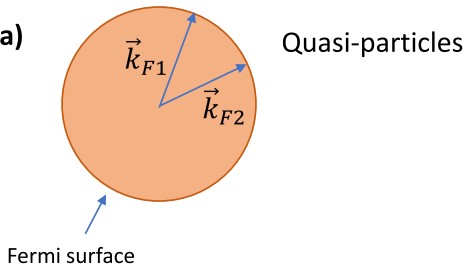

$$\frac{\kappa}{T}\Big|_{qp} = \frac{2\pi}{9}\frac{k_B^2}{h}k_F^2\ell_{qp} \propto T^{-2}$$

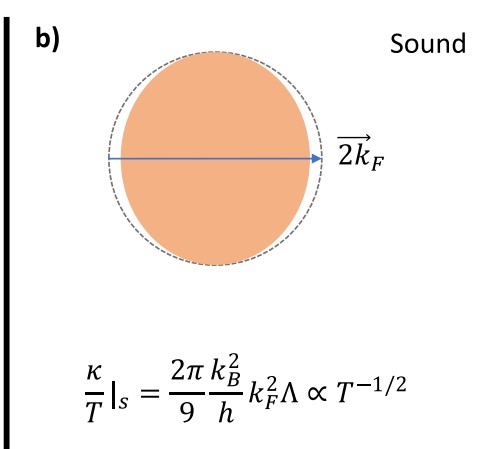

$$\frac{\kappa}{T}\Big|_{s} = \frac{2\pi}{9}\frac{k_B^2}{h}k_F^2\Lambda \propto T^{-1/2}$$

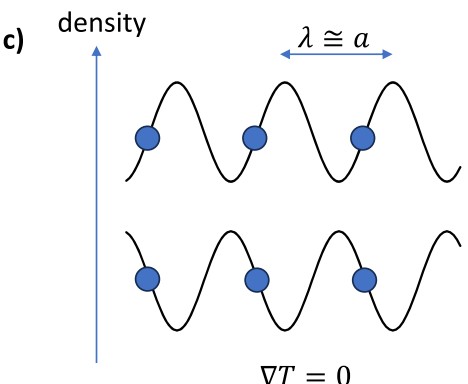

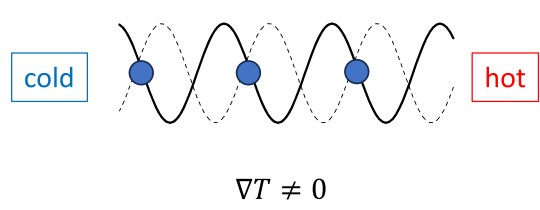

**Fig. 5 | Two channels for the conduction of heat. a** Transmission by quasi-particles is dominant at low temperatures. Collisions in momentum space between quasi-particles lead to a $\propto T^{-2}$ transmission. **b** Transmission by collective response of the Fermi surface. The amplitude of transmission, set by the square of the Fermi radius and the de Broglie thermal wavelength, the thermal thickness of the Fermi surface, is $\propto T^{-1/2}$. **c** Sound mode with a $2k_F$ wavevector has a wavelength of the order of the interatomic distance. A thermal gradient will cancel the equivalency between two modes propagating in opposite orientations.

atoms. A thermal gradient lifts this degeneracy. The fundamental reason behind the success of this simple approach is yet to be rigorously understood.

After this paper was written, we learned of two relevant early works, by Brazovskii[44] on the role played by a soft mode in the crystallization of a liquid, and by Dyugaev[45] on $2k_F$ rotons in $^3$He.

Arguably, normal liquid $^3$He is the cleanest and the simplest known Fermi liquid. If collective excitations play such a central role in its transport properties across such a wide temperature range, what about other strongly interacting systems of fermions residing beyond Landau's paradigm[46]? We leave this question to future studies.

## Methods

The thermal conductivity data published in references[17,31] were obtained using standard one-heater-two-thermometer set-ups either with a dilution[17] or a $^4$He[31] refrigerator. Measuring the thermal conductivity of Cu70-Ni30, Greywall[17] estimated that the precision of the data obtained by his cell was a few tenths of a percent. Horst and Meyer[31] discarded possible experimental errors emanating from convection by performing measurements at different temperature gradients.

## Data availability

All data generated during this study are included in this paper. The experimental data supporting the findings of this study are taken from Fig. 6 in ref. [17] and from Fig. 1 in ref. [31].

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

## Acknowledgements

K.B. acknowledges discussions with Mikhail Feigelman and is supported by the Agence Nationale de la Recherche (ANR-19-CE30-0014-04) and by the National Science Foundation (under Grant No. NSF PHY11-25915). K. T. is grateful to EPSRC for its support.

## Author contributions

The main idea of the paper was born during the discussions between the two authors. K.B. wrote the text with continuous feedback from K.T.

## Competing interests

The authors declare to have no competing interests.
