## [Peer Review File · Nature Communications]

REVIEWER COMMENTS

Reviewer #1 (Remarks to the Author):

The manuscript by Behnia and Trachenko discusses the "non-Fermi liquid (NFL)" property of heat transport in normal ^3He . In the standard Fermi liquid theory, the thermal conductivity is expressed as a function of the heat capacity and quasiparticle scattering time, resulting in the characteristic temperature dependence of $1/T$. At extremely low temperatures, the properties of the normal ^3He are described with the Fermi liquid theory. When the temperature increases by approximately 100mK, however, the Fermi liquid picture no longer applies. In the manuscript, the authors start by examining the thermal conductivity measurements by Greywall, which show a deviation from the typical FL behavior and a crossover to $T^{1/2}$ behavior expected in the gas-like regime. They associate the characteristic temperature at which the thermal conductivity exhibits NFL behavior with the Planckian time. Above the temperature, thermally excited sound modes come into play as the leading carriers of heat, which explains the $T^{1/2}$ behavior of the thermal conductivity in the high-temperature regime.

In the manuscript, the authors present a scenario to explain the crossover between the FL and NFL regions of the normal ^3He and shed light on the impact of collective excitations in heat transport. In terms of substance, I recommend its publication in Nature Communications after clarifying the following points.

1. To explain the $T^{1/2}$ behavior at high temperatures, the authors identify the sound modes as "a quantum mechanical wave" and apply the Landauer formula. Can you clarify the temperature range in which this theory applies? For instance, it would be helpful to know if there is a maximum temperature limit for this theory.
2. How important is the effect of the coupling of sound modes with quasiparticles? I think Eqs.(1)-(3) explain the temperature dependence of the thermal conductivity in the deep NFL regime (namely, the much higher temperature regime that the fermion-fermion scattering time becomes comparable to the Planckian time). In the crossover region (e.g., around 100mK at zero pressure), does the mutual coupling between quasiparticles and collective modes have a significant impact on the deviation of the thermal conductivity from both $1/T$ and $T^{1/2}$ behaviors?
3. It would be useful to explicitly mention the values of the fitting parameters " a_2 " and " $a_{1/2}$ " (Fig.4a) in the manuscript.

4. It has been discussed that the coupling of quasiparticles to zero sound phonons gives a " $T^3 \log(T)$ " behavior for the specific heat, but the magnitude is too small to account for the experimental observation. However, the spin fluctuation significantly contributes to the specific heat, resulting in the right order of magnitude to account for the experimental data of the specific heat [e.g., W. F. Brinkman and S. Engelsberg, Phys. Rev. 169, 417 (1968) and D. Coffey and C. J. Pethick, Phys. Rev. B 37, 1647 (1988)]. Spin fluctuations also give rise to the leading-order corrections to the FL behavior of the thermal conductivity [e.g., M. J. Rice, Phys. Rev. 159, 153 (1967) and K. S. Dy and C. J. Pethick, Phys. Rev. 185, 373 (1969)]. Do spin fluctuations have a significant impact on heat transport even in the deep NFL regime? Can you have any discussions and comments about this?

5. Minor typos: In Eq.(3), I think " m " should be " m^{\ast} ". There are several repeated twice words in the manuscript, e.g., "of of" in the 4th paragraph of page 1 and "decreases decreases" in the 3rd paragraph of page 3.

"How heat propagates in 'non-Fermi liquid' ^3He " Behnia and Trachenko

Liquid He-3 is the fundamental quantum liquid of great importance for our understanding of other superfluid and correlated states. It has been at the forefront of low-temperature physics for decades and still provides us with many unanswered puzzles.

At low temperatures, below than 10-100 mK its behavior generally follows the laws set out by the Fermi-Liquid paradigm, where main players are weakly-interacting quasiparticles. At higher temperatures, the quasiparticle picture breaks down and more complicated collective particle dynamics becomes dominant.

Heat transport and its temperature evolution from $1/T$ (low T) to $T^{1/2}$ (high T) is one of the unanswered questions. The manuscript proposes a qualitative picture of the heat transport in liquid He-3 at temperatures beyond the Fermi Liquid regime. The authors suggest that a density fluctuation mode, or sound, is the main carrier of energy in the system, based on the fact that the typical relaxation, or scattering time, falls below the Planckian time. They provide a simple model for the thermal conductivity based on Landauer transport formulation. It is a new approach to the problem, that may open new fruitful investigation directions.

Overall, the paper is nicely written, with relevant references and supplementary materials provided. The presented idea is interesting but I'd like the authors to clarify several points, and make the arguments more quantitative:

1) The Landauer transport formula (1) uses transmission coefficient in the form $q^2 * \ell$ for each mode. What are the applicability limits of this expression? Does it work for both ballistic quasiparticle propagation and the overdamped waves? Could authors provide a more detailed derivation/explanation of this?

2) The logical step on page 3, between avoiding "Debye sphere" T^3 dependence by involving transmission by the whole Fermi surface is not clear. Why would this be the case?

3) What are the numerical values of the fitting coefficients a_2 and $a_{1/2}$ in Fig 4a? Do they agree with the predictions given by the Landauer formula? The high-temperature behavior of heat conductivity (3) is shown in Fig 1a. Please add a similar asymptotic expression for the low temperature Fermi-Liquid regime.

4) Some of the qualitative arguments are not sufficiently supported. For example, why the de Broglie thermal length is the appropriate mean free path for the collective sound mode? Why the particular mode selected is the $2k_F$ mode? In the

"quantum" Bridgman formula (3) the speed of sound replaced by the thermal velocity, why not the group speed of the $2k_F$ mode?

5) The sound mode that is proposed for the heat transport is the " $2k_F$ hydrodynamic sound mode". This sound mode would have wavelength on the order of interatomic distance, with large attenuation. Why is this short wavelength excitation still a hydrodynamic regime "sound"? Does the Landauer formula apply here at all? Authors should provide more details on that. There are questions in the literature about how low in energy this mode really goes, according to references [47,48,51]. The question is whether this mode can be sufficiently thermally excited at 1K (0.1meV)? Possibly, the particle number n in (3) to be used is a small fraction of the total concentration? Moreover, the schematic depiction of the mode in Fig 4c is not clear. It looks more like a single particle-hole excitation at $2k_F$ momentum?

6) What about other collective modes? For example, deviations of the heat capacity and (to a lesser degree) thermal conductivity from Fermi-Liquid predictions at low temperatures has been explained by spin fluctuation in the paramagnon model, e.g. by Mishra, Ramakrishnan in Phys. Rev. B 31, 2825 and overviewed in monographs on He-3 by Dobbs, and by Vollhardt and Wolfle. Do other collective modes contribute to the heat transport?

7) additionally:

- a) in figures, the negative sign in powers of T merges with the top bar in " T " and hard to read;
- b) in Table I it is not clear what is meant by "Driver" in column 4;
- c) the summary paragraph is very brief and does not elaborate or expand on the main message of the paper.

Reply to reviewers

We thank both reviewers for the time they devoted to our manuscript, their careful reading and their insightful remarks. The two reports have helped us to improve the clarity of our message. Here is our point-to-point answer to their comments.

Reviewer #1 :

The manuscript by Behnia and Trachenko discusses the "non-Fermi liquid (NFL)" property of heat transport in normal ^3He . In the standard Fermi liquid theory, the thermal conductivity is expressed as a function of the heat capacity and quasiparticle scattering time, resulting in the characteristic temperature dependence of $1/T$. At extremely low temperatures, the properties of the normal ^3He are described with the Fermi liquid theory. When the temperature increases by approximately 100mK, however, the Fermi liquid picture no longer applies. In the manuscript, the authors start by examining the thermal conductivity measurements by Greywall, which show a deviation from the typical FL behavior and a crossover to $T^{1/2}$ behavior expected in the gas-like regime. They associate the characteristic temperature at which the thermal conductivity exhibits NFL behavior with the Planckian time. Above the temperature, thermally excited sound modes come into play as the leading carriers of heat, which explains the $T^{1/2}$ behavior of the thermal conductivity in the high-temperature regime.

In the manuscript, the authors present a scenario to explain the crossover between the FL and NFL regions of the normal ^3He and shed light on the impact of collective excitations in heat transport.

We thank the reviewer for this accurate summary of the context of our work and its significance.

In terms of substance, I recommend its publication in Nature Communications after clarifying the following points.

1. To explain the $T^{1/2}$ behavior at high temperatures, the authors identify the sound modes as "a quantum mechanical wave" and apply the Landauer formula. Can you clarify the temperature range in which this theory applies? For instance, it would be helpful to know if there is a maximum temperature limit for this theory.

Yes, it is true that we should have clarified this. We assume that the temperature is well below the Debye temperature of the system, which is ~ 20 K in solid ^3He (and slightly lower in more dilute liquid ^3He). Moreover, we are restricted upward by the degeneracy temperature of ^3He , which is ~ 5 K for non-interaction fermions and ~ 2 K for interacting fermions. The difference between the two Fermi energies is a result of mass normalization ($m^* \sim 2.9 m_3$).

2. How important is the effect of the coupling of sound modes with quasiparticles? I think Eqs.(1)-(3) explain the temperature dependence of the thermal conductivity in the deep NFL regime (namely, the much higher temperature regime that the fermion-fermion scattering

time becomes comparable to the Planckian time). In the crossover region (e.g., around 100mK at zero pressure), does the mutual coupling between quasiparticles and collective modes have a significant impact on the deviation of the thermal conductivity from both $1/T$ and $T^{1/2}$ behaviors?

We are indeed neglecting any coupling between quasi-particles and sound. This does not mean that there is no such coupling, only that one does not need to invoke it to explain the thermal conductivity data. Let us summarize the situation:

- i) If one assumes that heat is **only** carried by quasi-particles, one expects $\kappa \propto T^{-1}$. This only works when $T < 0.01$ K.
- ii) If one assumes that heat is **only** carried by a sound mode (with $q=2k_F$), one expects $\kappa \propto T^{1/2}$. This only works when 0.7 K $T < 2$ K.
- iii) If one assumes that heat is carried by both quasi-particles and sounds, neglecting any interaction between them, one can explain the experimental data with an accuracy better than 10 percent from 0.008 K to 2 K (See Fig. 4b in the new version).

In the new version, we have specified that we are neglecting possible coupling between the two conducting channels and this may be one of the reasons of the slight disagreement.

3. It would be useful to explicitly mention the values of the fitting parameters " a_2 " and " $a_{1/2}$ " (Fig.4a) in the manuscript.

Yes! We have provided this information explicitly in the new version (See table 2).

4. It has been discussed that the coupling of quasiparticles to zero sound phonons gives a " $T^3 \log(T)$ " behavior for the specific heat, but the magnitude is too small to account for the experimental observation. However, the spin fluctuation significantly contributes to the specific heat, resulting in the right order of magnitude to account for the experimental data of the specific heat [e.g., W. F. Brinkman and S. Engelsberg, Phys. Rev. 169, 417 (1968) and D. Coffey and C. J. Pethick, Phys. Rev. B 37, 1647 (1988)]. Spin fluctuations also give rise to the leading-order corrections to the FL behavior of the thermal conductivity [e.g., M. J. Rice, Phys. Rev. 159, 153 (1967) and K. S. Dy and C. J. Pethick, Phys. Rev. 185, 373 (1969)]. Do spin fluctuations have a significant impact on heat transport even in the deep NFL regime? Can you have any discussions and comments about this?

The answer to the first question is negative. There is no detectable evidence of a contribution by spin fluctuations (either as conductors of heat or as scattering centers) in the heat transport data when $T > 0.1$ K. Let us recall that the nuclear exchange energy is in the range of 1 to 2 mK, orders of magnitude smaller than the two other relevant energy scales (Fermi and Debye).

We examined the first two papers mentioned by the reviewer. Indeed, the first two papers propose a contribution to specific heat by spin fluctuations. However, this is restricted to low temperature. (below 0.04 K in the case of Brinkman & Engelsberg, and below 0.1 K in the case of Coffey and Pethick). As for the papers on thermal conductivity, Dy and Pethick explicitly focus on what they call "extreme low temperature limit" and do not attempt to explain the experimental data at finite temperature.

The paper by M. J. Rice stands out. He explicitly invokes scattering by spin fluctuations in order to explain the upward deviation of the thermal conductivity from T^{-1} up to 0.4 K. Therefore, his scenario is an alternative to ours. However, he has assumed an energy scale for spin fluctuations as large as 1.1 K. No justification is given. This is orders of magnitude larger than the nuclear spin exchange energy quantified by experiments. Indeed, nuclear magnetic relaxation measurements have found that $J=0.002$ K when the molar volume is $24.6 \text{ cm}^3/\text{mol}$ (Meyer, J. Appl. Phys. 39, 390–396 (1968)) and specific heat measurements find that $J=0.00085$ K, when the molar volume is $24.45 \text{ cm}^3/\text{mol}$ (). While there is a twofold discrepancy between these two measurements, they are respectively 550 times and 1300 times smaller than what is assumed by Rice.

Besides the unrealistic energy scale, there is a second objection to Rice's scenario. The deviation of thermal conductivity from its expected behavior is *upward*. Why then invoke an additional *scattering mechanism* (as he does) instead of an additional *conducting channel* (as we do)?

We have added a new discussion detailing all this in the Supplemental material .

5. Minor typos: In Eq.(3), I think "m" should be " m^{\ast} ". There are several repeated twice words in the manuscript, e.g., "of of" in the 4th paragraph of page 1 and "decreases decreases" in the 3rd paragraph of page 3.

Thanks for this careful reading. We have corrected them in the new version.

Reviewer #2 :

"How heat propagates in 'non-Fermi liquid' ^3He " Behnia and Trachenko

Liquid He-3 is the fundamental quantum liquid of great importance for our understanding of other superfluid and correlated states. It has been at the forefront of low-temperature physics for decades and still provides us with many unanswered puzzles.

At low temperatures, below than 10-100 mK its behavior generally follows the laws set out by the Fermi-Liquid paradigm, where main players are weakly-interacting quasiparticles. At higher temperatures, the quasiparticle picture breaks down and more complicated collective particle dynamics becomes dominant.

Heat transport and its temperature evolution from $1/T$ (low T) to $T^{1/2}$ (high T) is one of the unanswered questions. The manuscript proposes a qualitative picture of the heat transport in liquid He-3 at temperatures beyond the Fermi Liquid regime. The authors suggest that a density fluctuation mode, or sound, is the main carrier of energy in the system, based on the fact that the typical relaxation, or scattering time, falls below the Planckian time. They provide a simple model for the thermal conductivity based on Landauer transport formulation. It is a new approach to the problem, that may open new fruitful investigation directions.

Overall, the paper is nicely written, with relevant references and supplementary materials provided. The presented idea is interesting but I'd like the authors to clarify several points, and make the arguments more quantitative:

Many thanks for this nice summary and this positive assessment.

1) The Landauer transport formula (1) uses transmission coefficient in the form $T = \sum_{\ell} T_{\ell}$ for each mode. What are the applicability limits of this expression? Does it work for both ballistic quasiparticle propagation and the overdamped waves? Could authors provide a more detailed derivation/explanation of this?

The Landauer's approach to conduction is to consider the transmission of a wave, which is attenuated over a distance. When the wave is a ballistic quasi-particle, the attenuation distance (or the mean-free-path) is set by the finite size. One virtue of this approach is the transparency of the presence of fundamental constants and specific context-dependent length scales in setting the expected magnitude and temperature dependence of a transport coefficient. Here are two examples:

a) Thermal and electrical conductivity in a metal

The Landauer formalism leads to the following expression for the electrical conductivity of a two-dimensional metal whose Fermi surface is a circle of radius k_F and its mean-free-path ℓ :

$$\sigma^{2D} = \frac{e^2}{h} k_F \ell$$

This is because the number of conducting mode is $2 \times \frac{1}{2} \times 2\pi/\lambda_F$ (2 for spin degeneracy, $\frac{1}{2}$ for averaging a vector in two dimensions, and 2π is the total planar angular range). In three dimensions, one finds:

$$\sigma^{3D} = \frac{2}{3\pi} \frac{e^2}{h} k_F^2 \ell$$

Here, the Fermi surface is a sphere of radius k_F and there are $2 \times \frac{1}{3} \times 4\pi/\lambda_F^2$ conducting modes.

Replacing the Fermi radius by the carrier concentration (in 2D, $k_F^2 = 2\pi n$ and in 3D, $k_F^3 = 3\pi^2 n$) and the mean-free-path by the scattering time ($\ell = \tau \frac{\hbar k_F}{m^*}$) transform both these expressions to the familiar Drude expression:

$$\sigma^{Drude} = \frac{ne^2\tau}{m^*}$$

The Landauer expression for thermal conductivity in three dimensions, in agreement with the Wiedemann-Franz law, is:

$$\frac{\kappa^{3D}}{T} = \frac{\pi^2}{3} \sigma^{3D} = \frac{2\pi}{9} \frac{k_B^2}{h} k_F^2 \ell$$

This is a specific case of Equation 1 of our paper where the transmission coefficient is $\frac{2}{3\pi} k_F^2 \ell$.

b) Thermal conductivity of phonons in an insulator

Consider now an insulating cubic crystal with a lattice parameter equal to a . The phonon thermal conductivity at low temperature according to the Landauer formalism is:

$$\frac{\kappa^{ph}}{T} = \frac{\pi^2 k_B^2}{3 h} \mathcal{J}^{ph}$$

To quantify \mathcal{J}^{ph} let us assume that we are at low temperature, in the Casimir limit, where the mean-free-path of all phonons, is the size of the crystal L . The number of conducting modes is roughly: $\frac{1}{3} \times 4\pi q_D^2$. The latter is the external area of the Debye sphere. This leads us to:

$$\frac{\kappa^{ph}}{T} = \frac{4\pi^3 k_B^2}{9 h} q_D^2 L$$

Replacing $q_D = \frac{T}{a\Theta_D}$, where a is the lattice parameter and Θ_D the temperature. This leads to:

$$\kappa^{ph} = \frac{4\pi^3 k_B^2}{9 h} T \left(\frac{T}{a\Theta_D} \right)^2 L$$

The sound velocity can be introduced using: $v_s = ak_B\Theta_D/\hbar$ and the atomic density using $n = a^{-3}$

$$\kappa^{ph} = \frac{8\pi^4}{9} n \left(\frac{T}{\Theta_D} \right)^3 v_s L$$

One can see that this equivalent to the familiar $\kappa = \frac{1}{3} C v \ell$ with specific heat given by its low temperature Debye expression. These are specific examples for validity of equation 1 of our paper. It implies that the amplitude of thermal transport coefficient in three dimensions is set by the quantum of thermal conductance, the wavelength square, the mean-free path and a geometric numerical factor.

The same approach can be used to quantify thermal conduction by sound in three dimensions. There are $\frac{2}{3\pi} k_F^2$ conducting modes, with a mean free path of Λ . Therefore:

$$\frac{\kappa_s}{T} = \frac{2\pi k_B^2}{9 h} k_F^2 \Lambda$$

2) The logical step on page 3, between avoiding "Debye sphere" T^3 dependence by involving transmission by the whole Fermi surface is not clear. Why would this be the case?

Heat transmitted by the acoustic phonons inside the Debye sphere leads indeed to $\kappa^{ph} \propto \left(\frac{T}{\Theta_D} \right)^3$ as detailed in our reply above. However, this contribution is negligibly small, because the Debye temperature is relatively large compared to our temperature window of interest. What we quantify is the transmission rate by a **single** mode with a temperature-independent wave-vector $q=2k_F$ and a temperature-dependent mean free path: $\ell = \Lambda \propto \sqrt{T}$. It happens that the amplitude and the temperature dependence found by this approach both match the experimental data.

3) What are the numerical values of the fitting coefficients a_2 and $a_{1/2}$ in Fig 4a? Do they agree with the predictions given by the Landauer formula? The high-temperature behavior of heat conductivity (3) is shown in Fig 1a. Please add a similar asymptotic expression for the low temperature Fermi Liquid regime.

We have specified these numbers in the new version. Yes, they do! Please see table 2 and Fig. 4 in the new version.

4) Some of the qualitative arguments are not sufficiently supported. For example, why the de Broglie thermal length is the appropriate mean free path for the collective sound mode?

The inverse of the de Broglie thermal length is a measure of the broadening of the Fermi-Dirac distribution around its k_F singularity. (See the figure below). To be more precise, the distance between the two extrema of the temperature derivative $\frac{\partial f}{\partial T}$ scale with $\frac{\lambda_F^2}{\Lambda^2}$. Therefore, it is a natural choice for measure of the distance over which a collective fermionic mode attenuates.

Why the particular mode selected is the $2k_F$ mode?

Our first answer is that by selecting this mode, one finds a reasonable account of the experimental data. A second answer is that in a classical liquid, as the Bridgman formula implies, the dominant heat carriers are phonons with large wave-vector $q = \frac{2\pi}{a}$ (a is the interatomic distance) and in our quantum liquid $2k_F = \frac{2}{a}(3\pi^2)^{1/3}$, which is close enough. A third answer is that the singularity of Lindhard function at $q = 2k_F$, which makes this wave-vector special. The fourth answer is that the experimental spectroscopic data (Fig. S2 in the supplement) finds a profound minimum near this wave-vector.

In the "quantum" Bridgman formula (3) the speed of sound replaced by the thermal velocity, why not the group speed of the $2k_F$ mode?

This is not an arbitrary choice. It comes out of the Landauer transmission approach. Let us note that the third law of thermodynamics requires entropy to vanish at zero temperature. We suspect that the thermal velocity is present in this quantum context because collective energy transport is impossible without a finite width in the Fermi-Dirac distribution.

5) The sound mode that is proposed for the heat transport is the "2k_F hydrodynamic sound mode". This sound mode would have wavelength on the order of interatomic distance, with large attenuation. Why is this short wavelength excitation still a hydrodynamic regime "sound"?

Yes! We agree that this sound mode has a wavelength of the order of interatomic distance (See Fig. 5c). Yes! The available experimental data obtained by inelastic neutron and X-ray scattering does not allow to say what is the minimal thermal energy to excite the mode at 2k_F. However, note that they could resolve a mode. Our assumption that this sound mode is present in our temperature of interest implies that it can be thermally excited even below 1 K. If this happens to be the case, then we are in the hydrodynamic limit, because $\omega \cong \frac{k_B T}{\hbar} > \tau^{-1}$. The latter inequality is a consequence of the scattering time becoming shorter than the Planckian time.

Does the Landauer formula apply here at all?

Whys not?

Authors should provide more details on that. There are questions in the literature about how low in energy this mode really goes, according to references [47,48,51]. The question is whether this mode can be sufficiently thermally excited at 1K (0.1meV)?

Yes! The reviewer is perfectly right. There has been a controversy about how to interpret the x-ray data near the minimum in the dispersion curve, which shows a roton-like minimum. The data was taken at 1.1 K and they could detect zero sound at Q=1.8 Å⁻¹ (2k_F) even at this temperature, but it was widely broadened. Our proposal should motivate a study to quantify the temperature dependence of the energy of the zero sound near this wave-vector.

Possibly, the particle number n in (3) to be used is a small fraction of the total concentration? Moreover, the schematic depiction of the mode in Fig 4c is not clear. It looks more like a single particle-hole excitation at 2k_F momentum?

The particle number n in Eq. 3 is derived from the radius of the Fermi sphere. Therefore, it represents the entire fermionic population. We have amended Fig. 4c to show that it represents a distortion of the Fermi surface by the sound mode. As the reviewer has noticed, it corresponds to an excitation with a wavelength as short as the interatomic distance. We have added a sketch to clarify this point.

6) What about other collective modes? For example, deviations of the heat capacity and (to a lesser degree) thermal conductivity from Fermi-Liquid predictions at low temperatures has been explained by spin fluctuation in the paramagnon model, e.g. by Mishra, Ramakrishnan in Phys. Rev. B 31, 2825 and overviewed in monographs on He-3 by Dobbs, and by Vollhardt and Wolfle. Do other collective modes contribute to the heat transport?

This is an excellent question, which was also raised by the other reviewer. Please see our detailed answer to comment 4 by the first reviewer.

7) additionally:

a) in figures, the negative sign in powers of T merges with the top bar in "T" and hard to read;

Thank you! It has been corrected.

b) in Table I it is not clear what is meant by "Driver" in column 4;

We replaced "driver" by "mechanism".

c) the summary paragraph is very brief and does not elaborate or expand on the main message of the paper.

You are right. We have extended it.

Summary of changes

- i) The abstract has been modified (remark #7c by reviewer 1).
- ii) A new figure (Fig. 5b) has been added and Fig. 4a in the previous version has become Fig. 4b). By explicitly showing both $\kappa(T)$ and κ/T (T) one can better appreciate how close, but still imperfect, is the agreement between the experimental data and the theory.
- iii) Fig. 4 has been modified to give a better depiction of the sound mode (remark #5 of reviewer 2).
- iv) The amplitude of the Debye temperature and the Fermi temperature have been explicitly mentioned (remark #1 by reviewer 1 and remark #2 by reviewer 1).
- v) The absence of coupling between the sound mode and the quasi-particle mode in our model has been mentioned and its possible consequence discussed (remark #2 by reviewer 1).
- vi) A table has been added explicitly listing the coefficients of the $a_2T^{-2}+a_{1/2}T^{-1/2}$ expression (remark #3 by reviewer 1 and remark #3 by reviewer 2).
- vii) Early works associating the non-Fermi liquid behavior with spin fluctuation is mentioned in the main text and a discussion of them is the subject of a new section in the supplement (remark #4 by reviewer 1 and remark #6 by reviewer 2).
- viii) Minor mistakes were corrected (remark #5 by reviewer 1 and remark #7 by reviewer 2).
- ix) The derivation of equations based on Landauer approach is clarified (remark #1 by reviewer 2).
- x) The title has become “How heat propagates in liquid ^3He ” to avoid jargon.

REVIEWERS' COMMENTS

Reviewer #1 (Remarks to the Author):

The authors have addressed all comments and questions I raised in my previous report, and they have made significant improvements to the manuscript. Therefore, I recommend that this paper be published in Nat. Commun. without any further changes.

Reviewer #2 (Remarks to the Author):

The authors provided detailed answer to the questions raised in the first round. The only remaining concern that I have is that equations (1-3) do not seem to appear in standard Solid state books, and require special knowledge, so they might be unfamiliar to He3 research community. Perhaps authors may consider adding an appendix to help the readers, briefly discussing the main points such as presence of mean free path in these equations, along the lines that authors used in their reply to referees.

Following the recommendation by reviewer 2, a new supplementary note (number 4) has been added in the Supplementary Information.